# Targeting NETosis in Acute Brain Injury: A Systematic Review of Preclinical and Clinical Evidence

**DOI:** 10.3390/cells13181553

**Published:** 2024-09-14

**Authors:** Marzia Savi, Fuhong Su, Elda Diletta Sterchele, Elisa Gouvêa Bogossian, Zoé Demailly, Marta Baggiani, Giuseppe Stefano Casu, Fabio Silvio Taccone

**Affiliations:** 1Department of Biomedical Sciences, Humanitas University, Pieve Emanuele, 20089 Milan, Italy; 2Department of Intensive Care, Erasme Hospital, Brussels University Hospital, Université Libre de Bruxelles, 1070 Brussels, Belgium; eldadiletta.sterchele@gmail.com (E.D.S.); elisagobog@gmail.com (E.G.B.); zoe.demailly@chu-rouen.fr (Z.D.); stefanocasu87@gmail.com (G.S.C.); fabio.taccone@ulb.be (F.S.T.); 3Laboratoire de Recherche Expérimentale des Soins Intensifs, Université Libre de Bruxelles, 1070 Brussels, Belgium; fuhong.su@ulb.be; 4Terapia Intensiva e del Dolore, Scuola di Anestesia Rianimazione, Università degli Studi di Milano, 20089 Milan, Italy; 5Medical Intensive Care Unit, CHU Rouen, Normandie Université, F-76000 Rouen, France; 6Neurological Intensive Care Unit, Fondazione Istituto di Ricovero e Cura a Carattere Scientifico San Gerardo dei Tintori, 20900 Monza, Italy; mb.baggiani@gmail.com

**Keywords:** neuroinflammation, NETosis, neutrophil extracellular traps, nucleosome, anti-histone therapy, acute brain injury

## Abstract

Acute brain injury (ABI) remains one of the leading causes of death and disability world-wide. Its treatment is challenging due to the heterogeneity of the mechanisms involved and the variability among individuals. This systematic review aims at evaluating the impact of anti-histone treatments on outcomes in ABI patients and experimental animals and defining the trend of nucleosome levels in biological samples post injury. We performed a search in Pubmed/Medline and Embase databases for randomized controlled trials and cohort studies involving humans or experimental settings with various causes of ABI. We formulated the search using the PICO method, considering ABI patients or animal models as population (P), comparing pharmacological and non-pharmacological therapy targeting the nucleosome as Intervention (I) to standard of care or no treatment as Control (C). The outcome (O) was mortality or functional outcome in experimental animals and patients affected by ABI undergoing anti-NET treatments. We identified 28 studies from 1246 articles, of which 7 were experimental studies and 21 were human clinical studies. Among these studies, only four assessed the effect of anti-NET therapy on circulating markers. Three of them were preclinical and reported better outcome in the interventional arm compared to the control arm. All the studies observed a significant reduction in circulating NET-derived products. NETosis could be a target for new treatments. Monitoring NET markers in blood and cerebrospinal fluid might predict mortality and long-term outcomes. However, longitudinal studies and randomized controlled trials are warranted to fully evaluate their potential, as current evidence is limited.

## 1. Introduction

Acute brain injury (ABI), including all types of stroke and traumatic brain injury, (TBI) is one of the leading causes of death and disability world-wide [1,2,3,4]. Given the limits of direct intervention on the primary injury once it has occurred, clinical focus is redirected towards the prevention of secondary injury [5], a comprehensive term encompassing all potential insults occurring later after the first insult. Secondary brain injury has been associated with dismal outcome in terms of mortality and long-term disability [6,7]. Among the determinants of secondary damage, inflammation is believed to have a pivotal role in promoting the propagation of injury. Several biomarkers have been investigated in preclinical models as well as TBI human population, with an increased interest towards leukocytes, including neutrophil recruitment in the injured site [8], neutrophil/lymphocyte ratio [9] and biomarkers released by neutrophils [10]. However, the significance of these markers is limited due to their low specificity for central nervous system injury, as they may also reflect systemic inflammation. Additionally, measuring neuroinflammation at the bedside is not straightforward. For instance, cerebrospinal fluid (CSF) analysis and positron emission tomography (PET) scan are potential methods, but they are not available in all the settings [11,12]. As such, there is growing interest in more specific and easily measurable blood markers, such as neutrophil extracellular traps (NETs). NETs are chromatin fibres decorated with antimicrobial proteins released by neutrophils in diverse inflammatory conditions aiming at neutralizing potential pathogenic stimuli. In physiological conditions, chromatin is located within the nucleus of eukaryotic cells. Its fundamental unit is the nucleosome, constituted by histone dimers and tetramers wrapped around DNA. Oligo-nucleosomes are efficiently degraded by blood endonucleases and metabolized in the liver or eliminated by immune cells.

NETs capture pathogens and enhance the inflammatory response to kill them, but in the case of excessive or dysregulated NETosis, this can exacerbate inflammation and further promote tissue injuries. NETs have been shown to be associated with increased severity of various diseases [13,14,15,16]. In acute brain injury (ABI), their presence was first discovered from brain samples; in patients dying from intracranial haemorrhage, post-mortem studies have detected the presence of NETs in brain specimens [17]. This has helped to define the two-wave patterns of neutrophil release (e.g., earlier, within the first 72 h after the event, and later, between 8 and 15 days) and the localization of neutrophils, namely within the hematoma and the surrounding brain tissue. Moreover, in acute ischemic stroke (AIS) NETs were suggested as an important determinant of *immunothrombosis*, e.g., the formation of thrombi that are mediated by ongoing inflammation and that contribute to ischemic lesions in humans [18].

As NETosis has been observed in translational models of ABI, researchers have started to consider it as a potential therapeutic target [19]. Many markers of NETs have been studied, such as cell-free DNA, nucleosomes as well as histones and their epigenetic modifications [17]. From this perspective, the pharmacological neutralization of histones has been investigated to reduce inflammation in several conditions, like sepsis or autoimmune diseases [20]. However, further investigation is warranted, as anti-histone therapy could also be a valuable therapy for ABI patients.

Therefore, the purpose of this study is to summarize all the relevant evidence regarding the potential association of plasmatic levels of NET markers with outcome intended as mortality and functional recovery in humans and animal models affected by ABI and to investigate the association of anti-histone therapy with the levels of circulating nucleosomes.

## 2. Materials and Methods

This is a systematic review of the current literature that aims to answer the following questions:When anti-histone therapy is compared with conventional treatments, is there any significant difference in terms of mortality, disability, and long-term prognosis in both animals and humans?Is there any significant variation in nucleosome levels in biological samples of ABI population compared to healthy controls?Is there any significant correlation between nucleosome levels and the severity of brain injury?

This review followed the Preferred Reporting Items for Systematic Reviews and Meta-Analyses (PRISMA) guidelines 2020 [21] and the protocol was registered on Prospero on 26 July 2024 with the registration number CRD42024547358.

### 2.1. Eligibility Criteria

Studies were selected according to the criteria outlined below.

#### 2.1.1. Study Designs

We included randomized controlled trials (RCTs), including cluster RCTs, controlled (non-randomized) clinical trials (CCTs) or cluster trials, prospective and retrospective comparative cohort studies, and case-control or nested case-control studies. We selected studies investigating humans and experimental animals with ABI that compared anti-histone therapy with conventional therapy or no specific treatment. We considered mortality, disability, long-term prognosis and severity of disease by validated severity scores as primary outcomes. Considering the scarce available evidence, small sample size was not an exclusion criterion.

#### 2.1.2. Participants

We included studies examining both experimental animals and adult humans with ABI. We used the Center for Disease Control and Prevention (CDC) definitions of TBI, aneurysmal subarachnoid haemorrhage (aSAH), acute ischemic stroke (AIS) and intracerebral haemorrhage (ICH) to include studies.

#### 2.1.3. Interventions

The interventions of interest were any therapeutic strategy, including pharmacological and non-pharmacological therapy targeted to the nucleosome. Among the pharmacological molecules, we considered all the relevant direct and indirect inhibitors of such histone release, pathway blockers and molecules neutralizing histone, molecules with an anti-inflammatory effect including scavenger molecules for reactive oxygen species (ROS).

#### 2.1.4. Comparators

We considered all the relevant comparisons involving interventions that address our research questions, including both standard of care and no intervention.

#### 2.1.5. Outcomes

Due to possible variation in disease definitions over time, we extracted definitions of outcomes as reported in individual studies. We extracted outcomes in all data forms (e.g., dichotomous, continuous) as reported in the included studies.

#### 2.1.6. Timing

There were no restrictions in terms of length of follow-up, for both humans and animals.

#### 2.1.7. Setting

There were no restrictions regarding the type of setting in studies involving humans. There were no restrictions concerning the type of setting in studies involving animals.

#### 2.1.8. Language

We exclusively included articles reported in English.

### 2.2. Search Strategy

Literature search strategies were developed using medical subject headings (MeSH) and text words related to acute brain injury. We searched through MEDLINE and EMBASE on 19 January 2024. The electronic database search was supplemented by searching for trial protocols through meta-Register (URL: Home|ClinicalTrials.gov, accessed on 19 January 2024). The literature search was limited to the English language, experimental animals, and human subjects. To ensure coverage of the literature, we reviewed the reference lists of included studies or relevant reviews identified through the search. We also consulted the authors’ personal files to make sure that all relevant material has been captured. Finally, we circulated a bibliography of the included articles to the systematic review team.

Both qualitative and quantitative studies were sought. No study design, date or language limits were imposed on the search, although only studies in English language were included. A draft of the research string to be used for search is hereby reported: “Traumatic brain injury” OR “Subarachnoid Haemorrhage” OR “Ischemic Stroke” OR “Intracerebral Haemorrhage” OR “Acute brain injury” AND (“histone” OR “nucleosome” OR “neutrophil extracellular trap” OR “cell free DNA”). Also, the term “neuroinflammation” was used as a second search criterion.

### 2.3. Study Records

#### 2.3.1. Data Management

Literature search results were uploaded to Rayyan.ai, an Artificial Intelligence (AI) tool for systematic literature reviews facilitating collaboration between reviewers and detection of possible duplicates. The team developed and tested screening questions and forms for level 1 and 2 evidence assessments based on the inclusion and exclusion criteria. Prior to the formal screening process, a calibration exercise was undertaken to pilot and refine the screening questions. Further, we provided training to new members of the review team not familiar with the AI tool and the content area prior to the start of the review.

#### 2.3.2. Data Collection Process

Three authors (SGC, MS, EDS) independently evaluated the titles and abstract of potentially eligible studies. Any disagreements on study eligibility or data extraction were resolved by discussion or by a senior researcher (FST).

#### 2.3.3. Outcomes and Prioritization

The primary outcome was the detection of any significant difference in terms of mortality or functional outcome in experimental animals and patients affected by ABI undergoing anti-NET treatments. The secondary outcome was the detection of any significant difference in the levels of circulating markers of NETosis in ABI animals and humans compared to healthy controls.

### 2.4. Risk of Bias in Individual Studies

To facilitate the assessment of possible risk of bias for each study, we collected information using different tools according to the nature of the studies; JBI Critical Appraisal Tool for cohort studies [22] was used to assess observational studies, implemented by reporting aside aspects borrowed by ROB2 Cochrane Collaboration tool, while ROBINS-I Cochrane Collaboration tool was used for human non-randomized interventional studies. For animal studies, we used the SYRCLE’S Risk of Bias tool, an adapted version of ROB2 tool for interventional studies on animals [23]. Clinical appraisal tools are reported in the Appendix A.

### 2.5. Data Synthesis

A systematic descriptive synthesis is provided with tables to summarize the characteristics and findings of the included studies. We extracted the following data both for studies on animals and humans when available: population, mechanism of injury, NET markers, methods, experiments, outcome and results. The descriptive synthesis explores relationships and findings both within and between the included studies. No statistical analysis or meta-analysis was performed due to the high heterogeneity among studies in terms of population, NETosis markers, and underlying pathology.

## 3. Results

Of a total of 1246 studies, we included 28 studies, of which 7 were conducted on experimental animals while 21 involved humans (Figure 1).

Preclinical studies mainly enrolled adult CD1 mice, an outbred strain characterized by genetic variability, C57BL/6 mice, an inbred strain characterized by genetic uniformity, or Sprague-Dawley (SD) rats, an outbred strain aged 8 to 10 weeks. One study [24] used female swine (40–50 kg) as animal model. Observations were collected for the first 24 h. Results on outcome were available for 195 animals overall; among them, 121 were in the interventional arm. A total of 2074 humans were enrolled overall in the selected clinical studies. All studies were prospective observational studies except a non-randomized control trial [25] conducted on acute ischemic stroke (AIS). Only 15 patients received anti-NET therapy. AIS was the most extensively studied disease in the spectrum of ABI (10 studies), followed by isolated TBI (9 studies) and SAH (2 studies). No studies were found that addressed other causes of ABI such as intraparenchymal hematoma or venous sinus thrombosis. For clarity, we reported the collected evidence by separating animal studies from human studies. Quality assessment for all the studies was reported among the Appendix A.

### 3.1. Preclinical Studies

Animal studies are summarized in Table 1. All included studies were monocentric non-randomized experiments. Blood samples, when specified, were collected from peripheral circulation [26] or central circulation, including retro-orbital sinus blood [27,28] and femoral artery [24]. Among preclinical studies exploiting Enzyme-linked immunosorbent assay (ELISA) technique [18,24,27], only Denorme et al. clearly reported catching and detecting antibodies while quantifying myeloperoxidase (MPO)-DNA complexes. Regarding the primary outcome, a limited number of studies compared anti-NET treatments with conventional treatments or placebo [18,24,26,27,29]. In these studies, the treated groups reported a consistent reduction in circulating NET markers levels compared to healthy controls. In a TBI swine model, Sillesen et al. reported a relevant association between nucleosome levels and markers of disease severity, such as lesion size and brain swelling. In this case, fresh frozen plasma seemed to have reduced circulating nucleosomes levels, with further attenuation of the radiological features of severity. Similar results were obtained in a study conducted by Denorme et al., in which mice submitted to anti-NET therapy with neonatal NET inhibition factor (nNIF) reported smaller brain infarct size at autopsy in comparison with mice treated with inactivated scrambled peptide (SCR). In this study, the nNIF arm reported increased survival at 7 days compared to SCR arm, as well as better functional outcome at 24 h and 21 days. In the study conducted by De Meyer et al., mice treated with recombinant DNAse I (rhDNAse I) showed a significant improvement in neurological outcome compared with the control group, as well as a marked reduction in infarct size. Wu et al. [29] investigated the effect of CD21, a novel phthalide with a dose-dependent neuroprotective effect on cerebral ischemia in rodents [30,31]. Kim et al. [26] reported that proinflammatory molecules likely induce an increase in circulating levels of cfDNA and neutrophil elastase (NE-DNA) complexes in AIS mice after in vitro stimulation. These findings highlighted the major role of P2X7R, a purinergic receptor known to be targeted by ATP, as well as its analogous BzATP. Specifically, they hypothesized that Adenosine TriphosPhate (ATP) may induce NETosis. They developed a murine model to test this hypothesis by first inducing NETosis with ATP. In another experimental arm, they stimulated NETosis administering the prototypic P2X7 receptor (P2X7R) agonist *BzATP*. Finally, they compared the first two models with a third one, where NET release was seemingly prevented by administering A804598, a selective P2X7R antagonist. In those post-ischemic mice, the release of cfDNA induced by ATP was significantly reduced when a P2X7R antagonist was administered simultaneously, although cfDNA level did not return to baseline. Moreover, citrullinated histone H3 (CitH3) production seemed to be markedly suppressed by treatment with apyrase, an enzyme hydrolysing ATP, but enhanced by co-treatment of BzATP, confirming ATP-P2X7R-mediated NETosis. We did not report this result in Table 1 as CitH3 was semi-quantitatively measured by immunoblotting. Regarding the secondary outcome, the totality of studies reported a significant increase of circulating NETosis markers in ABI mice compared to healthy controls or sham mice. Differences were recorded regarding the peak time of circulating markers. De Meyer et al. [27] registered the highest values of cfDNA and histone DNA complex at 24 h from initiation of a 2-h transient middle cerebral artery occlusion (tMCAO). Kmet’ova [28] reported a higher cfDNA level at 1 h in mice with TBI compared to the control group, decreasing afterwards with still a significant value at 2 h. Wang et al. [31] examined TBI mice undergoing repeated blasts; in the selected population, cfDNA peaked at 2 h after injury, followed by progressive decrease. Sillesen et al. [24] showed that in swine with TBI and haemorrhagic shock, nucleosome level reached the highest circulating concentration at 3 h after injury and subsequently decreased, while DNAse I level peaked immediately after event, with progressive reduction at 3 and 6 h from injury.

### 3.2. Clinical Studies

In total, 20 studies were prospective observational studies and 1 study was a non-randomized control trial. They are all displayed in Table 2. Only one study among those included [25] explored the effect of an anti-NETs drug, Edaravone Dexborneol in AIS patient but reported the levels of NET as primary endpoint. A significant increase in NETs markers was detected in all ABI patients compared to healthy controls [18,32,33,34].

Some studies found a positive correlation between levels of circulating NETs markers with unfavourable outcome or severity [33,35,36,37]. At admission, National Institutes of Health Stroke Scale (NIHSS) was the most used tool to objectively quantify the impairment caused by AIS patients [18,34,38], while TBI patients were stratified according to severity by Glasgow Coma Scale [33,35,39,40]. One of the two studies enrolling aSAH patients exploited Hunt and Hess classification as a grading system for severity [41]. The most used scale for long term outcome was the modified Rankin Scale (mRS) [18,34,38]. Two studies carried out in the same department established a threshold of 10,000 kilogenome eq/L of cfDNA which seemed to be associated with a higher probability of poor functional outcome at 3 months in AIS [34,38]. Interestingly, there was a significant correlation between higher initial cfDNA levels and fatal outcome after TBI [37], and between elevated number of cfDNA copies and lower GCS at admission and after 24 h [35]. In TBI, survivors showed a higher reduction in cfDNA in the first 24 h from injury compared to non-survivors [35]; in one study, a double top pattern of cfDNA in TBI at admission and at 72 h was registered. Moraes et al. sought to establish a cut-off level associated with mortality, finally set at 17,000 kilogenom eq/L with high specificity but low sensitivity; Campello et al. [37] had previously set a cut-off of 77,000 kilogenom eq/L for severe TBI patients, with slightly higher sensitivity but reduced specificity compared to Moraes’ analysis. When NETosis is sustained, it leads to a dysregulation of the actin scavenging system [42], therefore despite stationary or increased DNAse I level, its activity may be hampered. While DNAse activity does not seem to be significantly different between AIS patients with favourable neurological outcome and those with unfavourable outcome, the dysregulation of the host response could be a risk factor for stroke-associated infections, which were more frequent in AIS patients with lower DNAse I activity [36]. Zeng [41] and Witsch [43] were the only authors reporting quantitative measurement of NETs markers in the aSAH population. Higher levels of CitH3 [41] and MPO/DNA complex [43] were found in patients compared to controls. Specifically, Witsch compared levels of MPO-DNA complexes measured by ELISA technique in the subgroups developing vasospasm and delayed cerebral ischemia (DCI). While vasospasm seemed to be correlated with a reduction in circulating markers, DCI was associated with further increase. Zeng found a linear correlation between CitH3 and severity of SAH according to Hunt and Hess classification. Only one study [44] detected the potential association of CitH3 levels with radiological severity in AIS intended as white matter lesions (WML) on MRI after admission. Severity of WMLs was assessed by Fazekas scale, based on the visual assessment of FLAIR images in both periventricular and subcortical areas to which the examiner attributes a certain score according to the extent and the confluence of lesions. This tool was developed in 1987 by observing T2-weighted images in Alzheimer patients [45]. Increased CitH3 levels were independently associated with a greater WML burden in adjusted multivariable analysis.

**Table 2 cells-13-01553-t002:** Summary of included studies on human population. Acronyms: POC: prospective observational cohort; RCT: randomised controlled trial; AIS: acute ischemic stroke; aSAH: aneurysmal subarachnoid haemorrhage; TBI: traumatic brain injury; ab: antibody; PAD: protein-arginine deaminase; EdaB: Edaravone Dexborneol; MODS: multiorgan dysfunction syndrome.

Author	Study	Population	Mechanism of Injury	NET Markers	Methods	Interventions	Outcome	Results
Gao (2024) [46]	POC	Adults (n° = 90)	AIS	MPO-DNA complex NE-DNA complex CitH3- DNA complex Nucleosome	ELISA ELISA CitH3 ELISA kit (Jingmei Biotechnologies, China) human nucleosome ELISA kit (Jingkang Biotech, China)	Quantification of MPO-DNA, NE-DNA, CitH3-DNA and nucleosome in: -AIS (n° = 45) at hospital admission -healthy controls (n° = 45)	Difference in NET markers levels	Higher circulating marker levels in AIS (*p* < 0.0001)
Tiwari (2023) [34]	POC	Adults (n° = 188)	AIS admitted < 24 h from symptom onset	cfDNA	rtPCR (Quanti Nova probe, Qiagen, Germany) for β-globin gene (Qiagen-Rotor-Gene Q MDX, Qiagen, Germany)	Quantification of cfDNA in: -AIS (n° = 188) -healthy controls (missing number)	Association of circulating cfDNA with severity of AIS and poor outcome (mRS ≥ 3 at 3 months)	Levels of cfDNA >10,000 kilogenome eq/L associated with poor outcome (*p* 0.002); moderate predictive accuracy of cfDNA for stroke prognosis (AUC 0.74)
Zhang (2023) [44]	POC	Adults (n° = 322)	AIS	CitH3	ELISA	Quantification of CitH3 at admission in: -severe WMLs (n° = 148) -non severe WMLs (n° = 174)	Association of circulating CitH3 with severe WMLs (MRI Fazekas score ≥ 3)	Higher citH3 in patients with severe WMLs (45.2 ng/mL [17.8–82.6]) compared to control group (19.6 ng/mL [10.3–46.5]), *p* = 0.001
Denorme (2022) [18]	POC	Adults > 18 years old (n° = 54)	AIS admitted ≤ 48 h from symptom onset	CitH3 MPO-DNA complex DNAse I activity	CitH3 ELISA Kit (Cayman Chemical) In-house-made ELISA kit [C: anti-MPO antibody; D: anti-DNA antibody] DNase I assay kit (Abcam)	(A) Quantification of CitH3 and MPO-DNA complex in: -AIS (n° = 27) at H admission -healthy controls (n° = 27) (B) DNAse I activity in: -AIS (n° = 27) at H admission -healthy controls (n° = 27)	Association of NET markers with disability assessed by mRS at discharge	Both CitH3 (r = 0.45, *p* = 0.024) and MPO-DNA (r = 0.507, *p* = 0.01) as significant independent variable to predict stroke outcome
Grosse (2022) [36]	POC	Adults (n° = 184)	AIS undergoing thrombectomy	cfDNA DNase I Activity	Quant-iT™ PicoGreen^®^ assay (Invitrogen, Carlsbad, CA, USA) Fluorometric K429-100 DNase I Activity Assay Kit (BioVision, Milpitas, CA, USA)	Quantification of cfDNA and DNAse I activity before mechanical thrombectomy in: -AIS with unfavourable outcome (n° = 43) -AIS with favourable outcome (n° = 48)	Association of NET markers with long-term unfavourable outcome intended as mortality or disability (mRS = 3–6) at 3 months	Baseline cfDNA higher in patients with unfavourable outcome at 90 days (*p* = 0.044, AUC 0.623); cfDNA at 7 days was higher in those with unfavourable outcome at 90 days (*p*< 0.001, AUC 0.74); association of cfDNA at admission (*p* = 0.001, AUC 0.751) and at day 7 (*p* < 0.001, AUC 0.855) with mortality
Huang (2022) [25]	Non-RCT	adults, 18–85 years old (n° = 45)	AIS clinically diagnosed (NIHSS 3–20) admitted ≤ 48 h from symptom onset	MPO-DNA complex CitH3	Cell death detection ELISA kit (Roche, USA) ELISA (Cayman Chemical, USA)	Quantification of CitH3 and MPO-DNA at admission and at day 3 in: -AIS undergoing EdaB therapy (n° = 15) -AIS undergoing conventional treatment (n° = 15) -healthy donors (n° = 15)	Effect of Edaravone Dexborneol on NET release and BBB tight junction proteins (occludin)	After stroke onset: higher levels of MPO-DNA and citH3 in AIS (*p* < 0.01), no significant variability between conventional treatment and Eda.B group. At day 3: significant decrease in MPO-DNA, citH3 in Eda B group (*p*< 0.01). Strong correlation between NETs and serum occludin levels.
Cui (2020) [47]	POC	Adults (n° = 68)	AIS	cfDNA	BGISEQ-500 sequencer with BWA algorithm	Quantification of cfDNA in: -AIS (n° = 48) -healthy individuals (n° = 20)	Differences in NET markers levels	Higher circulating cfDNA in AIS (*p* < 0.05). Prevalence of larger fragments (300–400 bp) rather than smaller fragments (75–250 bp) in AIS.
Lim (2020) [48]	POC	Adults (n° = 81)	AIS	cfDNA DNA-histone complex	Quant-iT™ PicoGreen^®^ dsDNA kit (Molecular Probes) Cell Death Detection ELISAPLUS (Roche Diagnostics GmbH)	Quantification of cfDNA and DNA-histone in: -AIS (n° = 58) -healthy controls (n° = 23)	NET makers in AIS population and association of circulating cDNA levels with MACE incidence Define sensitivity of cDNA for AIS	Significant increase in circulating cfDNA in AIS, but not for DNA-histone. Higher incidence of MACE in AIS group but not statistically significant. High sensitivity of cfDNA for AIS (AUC 0.859).
Vajpeyee (2020) [38]	POC	Adults (n° = 54)	stroke-like symptoms with samples withdrawn ≤ 12 h from onset	cfDNA	rtPCR (Quanti Nova probe, Qiagen, Germany) n) for β-globin gene (Qiagen-Rotor-Gene Q MDX, Qiagen, Germany)	Quantification of cfDNA at admission in: -AIS with unfavourable outcome (missing number) -AIS with favourable outcome (missing number)	CfDNA as early marker of AIS severity and poor prognosis (mRS ≥3 at 3 months)	Higher circulating cfDNA in patients with poor outcome (*p* < 0.05). Favourable outcome in patients undergoing thrombolysis or mechanical thrombectomy (n° = 26), who presented cfDNA level <10,000 kilogenome-eq/L at admission (*p* < 0.05).
Valles (2017) [49]	POC	Adults (n° = 270)	AIS admitted ≤ 24 h from symptom onset	cfDNA Nucleoome CitH3	fluoroscopy with Sytox Green (Invitrogen, Carlsbad, CA, USA) Cell death detection ELISAPLUS (Roche Diagnostics, Madrid, Spain) Cell death detection ELISAPLUS (Roche Diagnostics, Madrid, Spain)	Quantification of cfDNA, nucleosome and CitH3 at admission in: -AIS (n° = 243) -healthy controls (n° = 27)	Differences in NETosis markers levels and their association with clinical characteristics, stroke severity and outcome at 1 year	Elevated circulating NET markers in AIS at admission; higher in those with NIHSS >14. At discharge, all the markers positively correlated with NIHSS score (≥6) and mRS score (>2); at multivariate analysis, citH3 was associated with AF and all-causes mortality at 1 year.
Cao (2023) [40]	POC	Adults (n° = 20)	Severe TBI at admission	cfDNA CitH3-DNA complex	Quant-iT PicoGreen dsDNA Assay kit (Invitrogen, USA) Cell Death ELISA (Roche, Basel, Switzerland)	Quantification of cfDNA and CitH3 at admission in: -TBI (n° = 10) -Healthy matched controls (n° = 10)	Association of NET markers with worse neurological function intended as low GCS and elevated ICP	Levels of cfDNA were significantly elevated in TBI; cfDNA inversely correlated to GCS and positively correlated to ICP.
Mi (2023) [50]	POC	Adults (n° = 16)	TBI	cfDNA	Quant-iT™ PicoGreen^®^ dsDNA assay kit (Invitrogen, Carlsbad, CA, USA)	Quantification of cfDNA at admission in: -TBI (n° = 8) -healthy control (n° = 8)	Differences in NETosis markers levels	Higher circulating cfDNA in TBI (*p* < 0.05)
Ben Zvi (2022) [51]	POC	Adults, 18–67 years old (n° = 60)	Isolated mild TBI with no pathological findings at head CT scan	cfDNA	Fast SYBR Gold assay + fluorescence 96-well fluorimeter (SpectraMax Paradigm, Molecular Devices)	(A) Quantification of cfDNA in: -m TBI at admission (n° = 30) -matched control (n° = 30) (B) Correlation of cfDNA at admission in: -Survivors with moderate/severe impairment (n° = 4) -Survivors with mild or no impairment (n° = 14)	cfDNA as a prognostic marker for post-concussion syndrome assessed by CCT 1 and CCT 2 at 3 months	Higher circulating cfDNA at admission in patients with moderate/severe cognitive impairment according to CTT 1
Hazeldine (2021) [42]	POC	Adults (n° = 155)	TBI	cfDNA DNase I Activity	fluoroscopy ELISA (LifeSpan BioSciences Inc., UK)	Quantification of cfDNA and DNAse I activity in: -Isolated TBI (n° = 21) TBI + extracranial injuries (n° = 53) -Extracranial injury (n° = 81) -Healthy control (n° = 75)	Association of cfDNA with MODS defined as SOFA score ≥ 6 for more than 2 days	Significantly higher circulating cfDNA in all trauma patients at all post-injury time points; at 48–72 h, higher cfDNA in TBI patients developing MODS (*p* < 0.05); in all patients, significant reduction of DNAse activity and increase in DNAse I at 1 h, 4–12 h, 48–72 h (*p* < 0.005)
Marcatti (2021) [32]	POC	Adults (n° = 93)	TBI requiring activation of Trauma Team	cfDNA	Quant-iT™ PicoGreen^®^ dsDNA assay kit (Invitrogen, Carlsbad, CA, USA)	Quantification of cfDNA at admission in: -sTBI (n° = 33) -mTBI (n° = 20) -healthy volunteers (n° = 20)	Association of cfDNA with TBI severity	Higher circulating cfDNA in TBI (*p* < 0.0001); most of cfDNA had mitochondrial origin.
Shaked (2014) [33]	POC	Adults (n° = 58)	Blunt isolated TBI admitted ≤ 4 h from event	cfDNA	fluoroscopy with fluorochrome SYBR Gold	(Quantification of cfDNA at admission in: -TBI (n° = 28) -Healthy controls (n° = 30)	Association of cfDNA with severity (GCS at admission), disability (GOS at hospital discharge) and mortality	cfDNA levels were significantly higher in non survivors, in severe TBI and in patients with disability
Moraes Rodrigues Filho (2014) [39]	POC	Adults > 16 years (n° = 213)	Severe TBI	cfDNA	rtPCR (Life Technologies, Carlsbad, CA, USA) for β-globin gene	(A) Quantification of cfDNA at 12 h from ICU admission in: -severe TBI (n° = 188) -healthy control (n° = 25) (B) Comparison of cfDNA levels at 12 h from ICU admission in: -non-survivors (n° = 66) -survivors (n° = 122)	Association of cfDNA and mortality after ICU admission	Higher circulating cfDNA as an independent variable associated with mortality (*p* <0.001): cut-off level of 171,381 kilogen eq/L measured 12 h after entry predicting fatal outcome with high accuracy (AUC 0.9). Correlation of higher DNA levels and lower GCS at admission (*p* = 0.001)
Macher (2012) [35]	POC	Adults (n° = 65)	Severe TBI	cfDNA	rtPCR	(A) Quantification of cfDNA in: -severe TBI (n° = 65) at admission, at day 1 and day 3 -healthy controls (missing number) (B) Quantification of cfDNA decrease ratio at 24 h, 48 h and 72 h in: -non-survivors (n° = 14) -survivors (n° = 51)	Differences in cfDNA markers levels and association with mortality	Higher cfDNA in TBI at admission, reduction at 24 h and 48 h and mild increase at 72 h; significant decrease of cfDNA within 24 h from injury in survivors. Correlation of cfDNA levels with GCS at admission and at 24 h (*p* < 0.05)
Campello Yurgel (2007) [37]	POC	Male adults (n° = 54)	Severe TBI admitted to ICU ≤ 24 h from injury	cfDNA	rtPCR for β-globin with Taqman system	(A) Quantification of cfDNA at admission and at 24 h in: -severe TBI (n° = 41) -healthy male volunteers (n° = 13)	cfDNA as a prognostic marker for mortality	Higher cfDNA at 24 h from admission significantly correlated to mortality (r = 0.356, *p* < 0.03); cut-off of 77,883.5 kilog-eq/L at 24 h predictive for fatal outcome (ROC curveAUC 0.71).
Witsch (2022) [43]	POC	Adults (n° = 78)	aSAH confirmed at CT scan	MPO-DNA complex	Cell Death Detection ELISA (Roche, Basel, Switzerland)	(A) Quantification of MPO-DNA complex in aSAH population (n° = 78) at admission and at day 4 (B) Comparison of MPO-DNA complex levels in: -aSAH with DCI (n° = 17) -aSAH without DCI (n° = 12) (C) Comparison of MPO-DNA complex levels in: -aSAH with VAS (n° = 17) -aSAH without VAS (n° = 12)	Association of MPO-DNA complex levels with DCI	Significant reduction in MPO-DNA levels at day 4 compared to baseline. Significant increase in MPO-DNA levels in DCI (*p* = 0.04) compared to those without DCI. Significant reduction in MPO-DNA levels in patients with vasospasm (*p* =0.006) compared to patients without vasospasm.
Zeng (2021) [41]	POC	Adults (n° = 20)	aneurysmal SAH ≤ 24 h from admission	CitH3	Cell death detection ELISA PLUS (Roche) [C: mouse anti-histone biotinylated antibodies-D: anti CitH3 antibodies]	Quantification of CitH3 at admission in: -aSAH (n° = 10) -healthy controls (n° = 10)	Association of CitH3 levels with aSAH severity assessed by Hunt and Hess classification	Significant increase of CitH3 in aSAH (*p* = 0.0012); linear correlation between plasma CitH3 and Hunt and Hess classification (*p* = 0.0118)

## 4. Discussion

### 4.1. Neuroinflammation in Acute Brain Injury

While the majority of preclinical studies compared the effect of anti-NET drugs with placebo [23,25,28,29], only one human study [24] compared the effect of an indirect anti-NETosis therapy with standard care. This study was further impacted by several bias risks and overall poor methodological quality. At present, few studies have been conducted on this topic, especially in humans. There is also a significant gap between the quantity and consistency of evidence in animal models and human populations.

The studies included in this review showed a clear trend toward an increase in NET markers after ABI. The correlation between higher cfDNA levels and fatal outcomes after injury [35,37,39] suggests that NETosis activation significantly impacts survival. Unfortunately, we are unable to interpret the increase in NET markers straightforwardly in this clinical context. Some authors have attempted to define a threshold beyond which circulating markers are associated with unfavourable outcomes, but these findings need validation in external cohorts [34,39]. Additionally, different diseases within the ABI spectrum exhibited varying patterns of increase in serum NET markers. A summary of the main findings as well as future directions for the therapeutic application of research on NETosis is illustrated in Figure 2.

However, several interesting aspects arise from this search. It is acknowledged that NETosis is not a rapid process; even the suicidal pathway takes 2–4 h to create DNA fragments decorated with antimicrobial proteins. This could explain the double-peak pattern observed in TBI [35]: immediately after the traumatic event, cfDNA may be released directly due to tissue damage, independently of NETosis. In the case of sustained inflammatory stimuli, NETosis occurs, and its degradation products are released accordingly.

In aSAH, early neutrophil infiltration within 10 min of rupture has been associated with an increased risk of vasospasm [52]. In the study by Provencio et al., neutrophil accumulation in the CSF of patients with aSAH was an independent predictor of delayed vasospasm. The activities of two key enzymes are increased in aSAH: MPO, associated with increased consumption of nitric oxide (NO), and NADPH oxidase, which promotes the release of ROS. This suggests that NETosis could be among the mechanisms promoting inflammation in SAH. In the brain tissue of experimental mice undergoing aSAH, not only was the neutrophil count significantly increased, but the levels of CitH3 and CD16/CD32, a marker of the proinflammatory subtype of microglia, were also significantly elevated at 24 h after injury. This encourages the production of proinflammatory cytokines, such as TNF-α and IL-1β. Switching off NETosis by administering a DNase I analogue reduced the expression of CD16/CD32 by preventing the transition of microglia towards a proinflammatory subtype. However, the evidence is insufficient, and the methodological quality is poor. Further studies are needed to assess the role of NETs in the development of vasospasm and DCI.

### 4.2. Potential Therapeutic Strategies

Animal models suggest that the early administration of anti-NETs drugs could significantly reduce the pro-inflammatory environment initiated by the primary lesion, thereby preventing secondary damage. Three classes of drugs are available for anti-histone treatment: inhibitors of histone release, signal pathway blockers, and molecules that neutralize histones. Other molecules targeting different NET components (cfDNA, NE) have been proposed in various settings, such as chemotherapy-resistant cancer [53].

Pharmacological inhibitors, such as PAD4 inhibitors (e.g., Cl-amidine), prevent the citrullination of histones, a crucial step in NET formation. DNase enzymes can degrade the DNA backbone of NETs, reducing their pro-inflammatory and cytotoxic effects. DNase I has been explored in preclinical studies for its potential to mitigate NET-related damage in conditions such as stroke and spinal cord injury. Several studies have tested the efficacy of anti-inflammatory agents, such as corticosteroids or specific cytokine inhibitors like IL-1β inhibitors, considering their indirect effect in reducing NETosis by dampening the overall microinflammatory environment [13,54].

Kim et al. [25] assessed the interplay between ATP as a pro-inflammatory damage-associated molecular pattern and NET markers. By neutralizing P2X7R, an ATP-gated cation channel involved in various immune responses including inflammasome activation and free radical production, the detrimental effects elicited by the ATP-P2X7R complex were arrested. They also tested Cl-amidine, an irreversible inhibitor of PAD enzymes. Denorme et al. [28] tested GSK199, a reversible yet potent inhibitor more specific to PAD4. As mentioned above, PAD isotypes are responsible for deamination, specifically the conversion of arginine residues of several proteins into citrulline, a process that increases the potential pro-inflammatory activity of histones. Results from murine models affected by type-1 diabetes [55] showed protective effects of this treatment against endothelial dysfunction and microvascular damage, especially in the kidneys.

Few interventional studies have been conducted on human populations. We retrieved only one study [24] presenting the results following administration of Edaravone Dexborneol to AIS patients. Edaravone acts as a scavenger for ROS [56] released after ischemic stroke. Through its mechanism of action, endothelial cell damage is inhibited, brain oedema is reduced, and delayed neuronal death is halted. Dexborneol acts as an anti-inflammatory drug by reducing the expression of pro-inflammatory cytokines such as IL-6 and facilitates the penetration of Edaravone through the blood–brain barrier, increasing the drug’s availability at the target site. Some studies explore the potential of DNase I administration in AIS [26] as well as in TBI [57] based on the pathophysiological evidence of its reduced activity. Ex vivo studies showed that thrombolysis seemed to be more successful when adding DNase I to standard t-PA, hypothesizing a synergistic effect [58]. Many commonly prescribed drugs have been hypothesized to exert an anti-NETosis effect; for instance, anti-platelets therapies are known to prevent the interaction between platelets and neutrophils. Moreover, immunomodulators such as cyclosporine A effectively inhibit the calcineurin pathway. Unfortunately, reliable evidence supporting the reduction of NET markers in humans affected by ABI is currently lacking. From the perspective of translating results from animal models to humans, we believe that disseminating evidence collected in animals can foster further human research.

### 4.3. Perspectives on Neuroprotection

Neuroprotection by modulating the release of NETs is an emerging area of research with considerable potential. NETosis represents a relatively novel and significant mechanism of immune response that plays a crucial role in the pathophysiology of various acute brain injuries. While other markers have been extensively studied, NETosis offers unique insights into the inflammatory processes, has the advantage of being easily measured at the bedside and has shown to be a promising therapeutic target. While NETs play a crucial role in trapping and killing pathogens, their dysregulated release and accumulation have been implicated in various neuroinflammatory and neurodegenerative conditions. Excessive NETosis has been linked to chronic neuroinflammation, contributing to the pathogenesis of neurodegenerative diseases such as multiple sclerosis and Alzheimer’s disease [45]. The components of NETs, particularly histones, are cytotoxic and can induce neuronal cell death, exacerbating neural damage. Therefore, modulating NET release could reduce the subacute or chronic inflammation underlying many conditions. Developing therapies that selectively target excessive NETosis without compromising the essential immune functions of neutrophils remains a critical challenge.

### 4.4. A Process Called NETosis

Since Brinkmann et al. (2004) [59] first described NETs, extensive research has focused on the antimicrobial activity of neutrophils. Neutrophils, part of the innate immune response, are recruited to infected tissues, engulf pathogens, and kill them using reactive oxygen species (ROS) and antimicrobial proteins in granules. Brinkmann et al. revealed that neutrophils release NETs, DNA fibres with histones and antimicrobial proteins, in inflammatory environments, a process distinct from necrosis and apoptosis.

There are two pathways of NETosis from nuclear DNA identified as vital or suicidal, distinguished primarily by the activation of NADPH oxidase 2 (Figure 3). The vital pathway is independent of this enzyme, and neutrophils remain alive for a while because the nucleus is gradually disaggregated into vesicles. The suicidal pathway is linked to the activation of NADPH oxidase 2, which occurs in the presence of an increased cytosolic concentration of calcium and is facilitated by the release of ROS by mitochondria. This is followed by the activation of the Raf-MEK-ERK pathway and the release of Neutrophil Elastase (NE) from azurophilic granules. NE translocate to the nucleus and cleaves the oligonucleosomes; afterward, histones undergo epigenetic citrullination of arginine residues by the enzyme peptidyl-arginine deiminase 4 (PAD4). The nuclear envelope undergoes substantial modifications leading to swelling and dissolution, followed by the release of NETs and simultaneous death of the cell.

Transmission electron microscopy (TEM) and immuno-microscopy have enabled close observation of the fascinating formation of NETs reproduced in vitro [60]. Once the pathogen has been neutralized, NET degradation products are dismantled by DNase I. NETs have been shown to have important effects in controlling bacterial [59] and viral infections [16]. Moreover, NETosis has been observed in a wide variety of diseases that share the creation of a proinflammatory microenvironment [61], ranging from cancer [62] to autoimmune conditions [63] and graft dysfunction [64]. Rubartelli et al. [65] presented an extensive overview of the major mechanisms of NETosis implicated in metabolic and infectious diseases.

At present, the most studied pathway is suicidal NETosis. Its effects, governed by a ROS-dependent death, exert such a potent action that they presumably extend beyond the neutrophil’s lifespan. If dysregulated, this beneficial process may foster a cascade of proinflammatory events, further promoting inflammation and tissue damage [66].

### 4.5. Markers

It is possible to detect several markers of NETosis in blood samples: cfDNA, citH3, NE-DNA complex, and MPO-DNA complex are the most studied molecules. CitH3 is a specific biomarker for the suicidal PAD4-dependent pathway of NETosis. Histones can undergo post-translational modifications; the enzyme peptidyl-arginine deiminase 4 (PAD4) is responsible for the conversion of arginine residues of histone H3 into citrulline, increasing the proinflammatory activity of the molecule [67]. Alongside CitH3 and histones, neutrophil elastase (NE) and myeloperoxidase (MPO) are two enzymes that translocate to the nuclear space after neutrophil activation. Similar to PAD4, they also promote chromatin decondensation and form extracellular traps composed of chromatin and proteases [68]. It is important to note that nucleosomes and circulating cell-free DNA can also be released from processes other than NETosis, such as apoptosis or necrosis; therefore, MPO/DNA, NE/DNA conjugates, and CitH3 may be more specific for NET analysis [63].

### 4.6. Different Techniques to Measure NETs Markers

Plasma or whole blood are the most commonly used substrates to measure NETs markers. Various techniques are available today. ELISA is an easy and cost-effective method to quantify CitH3, nucleosomes, MPO-DNA, and NE-DNA complexes. Depending on the selected marker, it can be more specific for NETosis, especially when targeting histones.

Fluorescence spectrophotometry can be utilized to quantitatively analyse NET formation thanks to ultra-sensitive nucleic fluorescent acids that do not penetrate impermeable cellular membranes, such as SYTOX Green and SYBR Gold. However, the main limitations of this method include the risk of false-positive results if cellular membrane integrity is compromised and interference with the binding of dyes to NETs markers in the presence of cationic molecules such as host defence peptides. Spectroscopy can be further optimized using PicoGreen (Invitrogen), an ultra-sensitive fluorescent nucleic stain for quantifying double-stranded DNA (dsDNA) in solution. Quantification of DNA released into the supernatant is possible after disaggregation of the NETs by the enzyme micrococcal nuclease. Results must be confirmed via microscopic observation to ensure that cfDNA is not released by necrotic cells. The main drawback of this assay is its low sensitivity when a limited number of neutrophils are releasing NETs.

More traditional flow cytometry techniques have several advantages, being automated, quantitative, and rapid. Some methods are combined with fluorescence microscopy to allow the quantification of NETs by neutrophils undergoing NETosis, with the limitation of missing cells in later stages of the process. Polymerase chain reaction (PCR) techniques can also be used to identify and quantify DNA extracted in real time [69]. The development of next-generation sequencing techniques has been boosted by the advent of epigenomics [70], allowing the identification of epigenetic histone modifications or the provenance of cfDNA using bioinformatic tools. Many other techniques could be used; for instance, many studies report Western blot as another method to detect NETs markers. Unfortunately, it only offers a semi-quantitative estimation; hence, we decided to discard all studies using this method from our research.

### 4.7. Limitations of the Study

This review has several limitations. Firstly, the overall quality of the studies considered was low, primarily due to missing data and lack of randomization in both animal and human cohorts. Some studies omitted the number of control patients. Most of the literature exploring these novel markers in the human ABI population consists of prospective cohort studies. Additionally, most of the human studies included were single-centred, and few of them had more than 100 participants, increasing the risk of small trial bias. Also, appraisal tools like ROBINS-I and SYRCLE’s have notable limitations, including subjectivity in assessment which can result in marked inter-rater variability. Specifically, SYRCLE’s tool for animal studies may face challenges due to incomplete reporting in preclinical research, complicating bias assessment. Both tools demand careful interpretation and a steep learning curve, which can affect their consistency and overall effectiveness.

Moreover, we acknowledge that the methods of sample processing, extraction, and identification of NETs degradation products have not been standardized. Due to the extreme heterogeneity of populations, markers, and outcomes, we could not perform any meta-analysis. Even among studies quantifying the same marker, the units of measurement often differed, making comparisons challenging. Additionally, the main sites of injury and the pathophysiological mechanisms behind ABI were not clearly stated in most clinical papers.

An important limitation of these markers, especially cfDNA, is the inability to assess their origin, i.e., brain damage or extracranial lesions, particularly in subjects with multisystemic trauma. Authors tried to overcome this potential confounding factor by including only patients with isolated TBI [33,51] or by comparing the levels of markers in patients with isolated TBI versus those with extracranial injuries [37,42].

We only included studies considering quantitative measurement of NETs markers and excluded all evidence of semi-quantitative detection of NETs markers. Regarding the stroke population, we excluded studies analysing only thrombi or post-mortem histological samples. Our choice was driven by the aim of clarity and pragmatism, to promote comparison and reproducibility of results. Nevertheless, it would be interesting to observe trends of NETs markers in other biological samples besides blood, such as CSF. We hypothesize that retrieving these markers in CSF could increase specificity for neurological damage. To our knowledge, there is no available evidence on acute brain-injured patients at present.

## 5. Conclusions

NETosis should be considered as a benchmark to define the intensity of neuroinflammation in acute brain injured patients and as a potential therapeutic target. Further longitudinal studies and RCTs are warranted to better understand their full potential, as evidence remains scarce.

## Figures and Tables

**Figure 1 cells-13-01553-f001:**
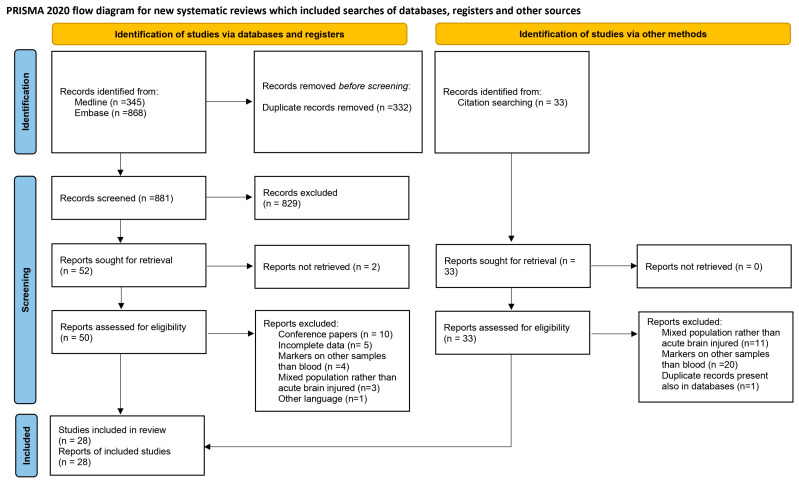
PRISMA flowchart of the review.

**Figure 2 cells-13-01553-f002:**
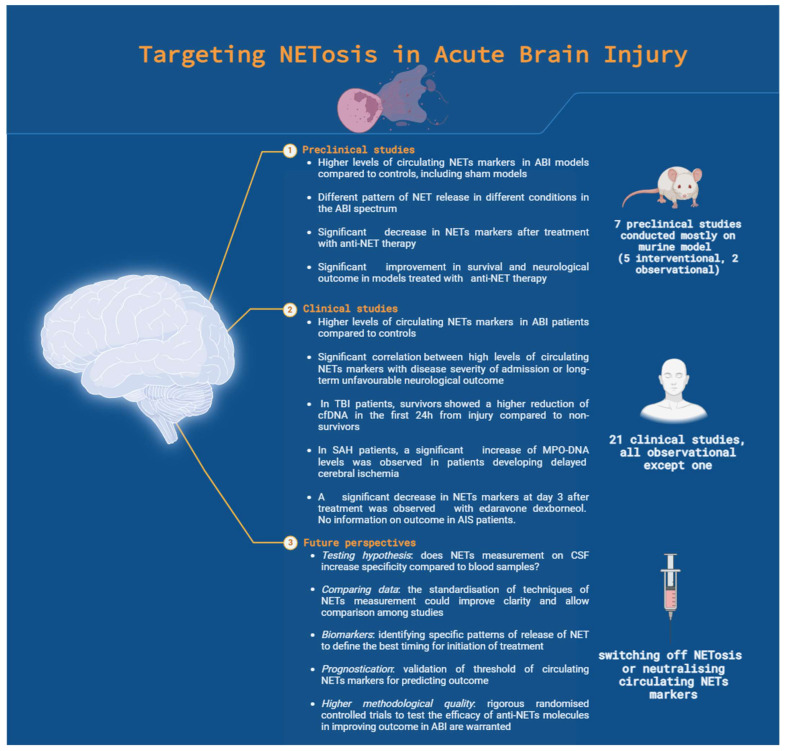
Summary of preclinical and clinical findings and future directions for the clinical application of research on NETosis (illustration created with BioRender.com). Acronyms: NET: neutrophil extracellular trap; ABI: acute brain injury; TBI: traumatic brain injury; SAH: subarachnoid aneurysmal haemorrhage; MPO-DNA: myeloperoxidase-deoxy-ribonuclease acid; CSF: cerebral spinal fluid.

**Figure 3 cells-13-01553-f003:**
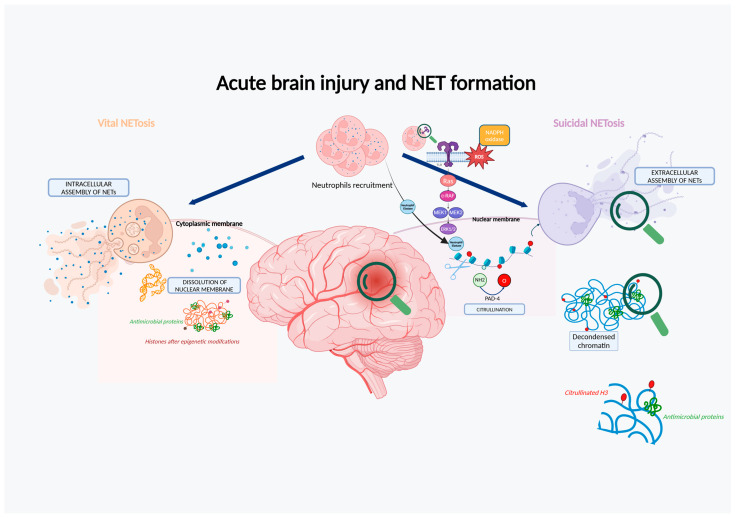
The main pathways of NET formation involved in neuroinflammation (illustration created with BioRender.com).

**Table 1 cells-13-01553-t001:** Summary of included studies on experimental models. POC: prospective observational cohort; INT: interventional; PTS: photothrombotic stroke; t MCAO: transient middle cerebral artery occlusion; p MCAO: surgical permanent right MCA occlusion; NIT: Neonatal NET-inhibitory factor; SCR: inactive scrambled peptide control; RhDNAse I: Recombinant Human DNAse I; RRT: righting reflex time; FFP: fresh frozen plasma; NS: normal saline.

Authors	Study	Population	Mechanism of Injury	NET Markers	Methods	Experiments	Outcome	Results
Wu (2023) [29]	INT	C57BL/6 mice	PTS	cfDNA	Quant-iT™ PicoGreen^®^ assay (Invitrogen, MA, USA)	Quantification cfDNA at 24 h after MCAO in: -mice injected with CD21 -mice injected with CD21 + pretreated with Compound C	Neuroprotective effect of CD21	CD21 significantly reduced levels of NETs in blood and ischemic brain tissues in PTS mice (*p* < 0.01); effect reversed by pretreatment with Compound C (*p* < 0.01).
Denorme (2022) [18]	INT	C57BL/6J mice 10–12-week-old male or female; 18-month-old male C57BL/6J mice	tMCAO	CitH3 MPO-DNA	Citrullinated Histone H3 ELISA Kit (Cayman Chemical) In-house made ELISA kit (C: anti-MPO antibody D: anti-DNA antibody)	(A) Quantification of MPO-DNA complex at 24 h after tMCAO in mice: -pretreated with BoxA (n° = 6) -pretreated with vehicle (n° = 5) (B) Quantification of MPO-DNA complex at 24 h after tMCAO in mice with DM I and older than 18 months: -injected with nNIF (n° = 10) -injected with SCR (n° = 9) (C) Quantification of MPO-DNA complex at 24 h after tMCAO in mice: -pretreated with GSK-199 (n° = 9) -pretreated with DNase I (n° = 9) -pretreated with vehicle (n° = 10)	Differences in NET markers levels, functional outcome assessed by modified Bederson test and Grip strength test and mortality on day 7 post t MCAO	nNIF mice reported better functional outcome (*p* < 0.01) and increased survival at 7 days (*p* = 0.0097). Treatment with nNIF BoxA DNAse I or GMKS significantly reduced MPO-DNA levels (*p* < 0.01).
Kim (2020) [26]	INT	Male Sprague-Dawley rats (8 weeks old, 230–250 g body weight)	pMCAO	cfDNA NE-DNA	Quant-iT™ PicoGreen^®^ dsDNA assay kit (Invitrogen, Carlsbad, CA, USA)	(A) Quantification of cfDNA in pMCAO mice: -treated with ATP (n° = 4) -treated with BzATP (n° = 6) -treated with PMA (missing number) -treated with ATP + A438079 (missing number) (B) Quantification of NE-DNA complex in PMCAO mice: -treated with ATP (n° = 4) -treated with ATP + A438079 (missing number) -treated with ATP + Cl-Amidine (missing number)	Release of NET markers after administration of ATP	Significant increase of circulating cfDNA upon injection of ATP (*p* < 0.05), BzATP (*p* < 0.01) or PMA (*p* < 0.01), reversed by A438079 (P2X7 antagonist) (*p* < 0.01). Significant increase of the release of NE-DNA induced by ATP (*p* < 0.01), reversed by A438079 (P2X7 antagonist) or CL-Amidine (*p* < 0.01).
De Meyer (2012) [27]	INT	Wild-type C57BL/6 8–10-week-old males	tMCAO	cfDNA Histone -DNA	fluorometric fluoroscopy (Fluoroskan; Thermo Fisher Scientific, Waltham, MA, USA) with fluorochrome Sytox Green Cell death detection ELISA or Cell Death Detection ELISA plus (Roche, Indianapolis, IN, USA)	(A) Quantification of cfDNA and histone-DNA complex in: -tMCAO mice at 24 h from the event (n° = 9) -sham mice (n° = 9) (B) Functional outcome assessment in: -mice treated with RhDNAse I (n° = 14) -mice treated with vehicle (n° = 15)	Early differences in NET markers levels, functional outcome assessed by modified Bederson test, grip test and corner test	Increase in cfDNA and histone-DNA complexes in AIS group baseline and after 24 h from stroke (*p*< 0.05). Better neurological outcome after treatment with RhDNAse I (*p* < 0.05).
Kmet’ova (2022) [28]	POC	young adult CD1 mice (n° = 44)	Blunt TBI	cfDNA *	QIAamp^®^ DNABlood MiniKit (Qiagen, Hilden, Germany) + QubitTM (Thermo Fisher Scientific, Waltham, MA, USA) + fluorometry	Quantification of cfDNA in: -TBI mice at 3 h from the event (n° = 34) -healthy control (n° = 10)	Early levels of cfDNA after TBI and its correlation with worse neurological function evaluated by static rods test at 1 month	Significant cfDNA increase in TBI after 1 h and 2 h (*p* < 0.05), Higher cfDNA in mice failing behavioural test (*p* = 0.014).
Wang (2014) [31]	POC	Male C57BL/6J mice (8–10 weeks old, weighed 22–26 g)	TBI from repeated blasts	cfDNA	fluorescence with SYBR Green I Nucleic Acid Gel Stain (Invitrogen Corporation, Grand Island, NY, USA)	Quantification of cfDNA at 2 h, 6 h, 24 h, 72 d in: -TBI mice (n° = 10) -sham mice (n° = 10)	cfDNA trend after repeated blast exposures and correlation with functional outcome assessed by RRT	Circulating cfDNA peaked 2 h after blast exposure (*p* <0.001); significant linear correlation between RRT and cfDNA (*p* < 0.005).
Sillesen (2013) [24]	INT	female Yorkshire swine (40–50 kg) (n° =12)	Blunt TBI + severe haemorrhagic shock	Nucleosomes DNAse I	Cell death detection ELISA plus, (Roche Applied Sciences, Indianapolis, IN, USA) ELISA (Bluegene Biotech, Shangai, China)	Quantification of nucleosomes and DNAse I in: -mice receiving FFP (n° = 6) -mice receiving NS (n° = 6)	Early differences in NET markers levels, correlation of circulating nucleosome and DNAse I with lesion size and brain swelling	Reduction in circulating nucleosome and DNAse I depletion after FFP; circulating nucleosome levels significantly correlated with lesion size (*p* = 0.002) and brain swelling (*p* < 0.001), circulating DNAse I correlated with brain swelling (*p* = 0.036) but not with lesion size (*p* = 0.124).

* = De Meyer et al. referred to cfDNA as extracellular DNA (ecDNA); Kim et al. referred to cfDNA as double-strand DNA (dsDNA).

## Data Availability

Data will be made available upon request to the corresponding author.

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
