# Peer review of "Targeting NETosis in Acute Brain Injury: A Systematic Review of Preclinical and Clinical Evidence"

_cells, 2024, doi:10.3390/cells13181553_

Round 1

Reviewer 1 Report

Comments and Suggestions for Authors

In the manuscript entitled ‘Targeting NETosis in Acute Brain Injury: a systematic review of preclinical and clinical evidence’, Savi and colleagues summarized both preclinical and clinical studies aiming at evaluating the impact of anti-histone treatments on outcomes in patients and animals with brain injuries. They discussed neuroinflammation in acute brain injury, potential therapeutic strategies, neuroprotection, NETosis and markers, as well as limitations of the study. Overall, the topic is important and the paper is well organized.

Author Response

Thank you for your valuable comments on this challenging topic. We have tried to make the most out of the available evidences.

Reviewer 2 Report

Comments and Suggestions for Authors

The present manuscript by Savi et al., titled "Targeting NETosis in Acute Brain Injury: a systematic review of preclinical and clinical evidence," aims to conduct a systematic review of the literature on the impact of NETosis in acute brain injuries, seeking to evaluate the efficacy of therapies both in preclinical models and human populations, focusing on outcomes such as mortality and functional recovery. However, some relevant points need to be revised, such as:

The objective is clear, but the introduction lacks a more robust justification for why NETosis was chosen as the therapeutic focus over other inflammatory processes. I also recommend improving the introduction by highlighting the specific importance of NETosis in acute brain injuries.

The description of study selection in the Materials and Methods section is well-described and adequate, but it lacks details about the exclusion of studies with smaller sample sizes (or if there was any sample size limit) or different methodologies. I suggest that these details be explained in the exclusion criteria and how they might impact the results.

The use of the SYRCLE and ROBINS-I tools was appropriate, but the manuscript does not discuss their limitations. I recommend including a brief discussion of these limitations to provide a more critical evaluation of the results.

The separation of preclinical and clinical results was clear, but the integration of the data could be improved. I suggest a greater integration of the findings, possibly with a comparative table or figure, thereby enhancing the visibility of the results for the reader and making the manuscript more appealing.

The discussion of limitations mentions the lack of randomized trials but does not sufficiently explore its impact. Therefore, I find it very important to include a paragraph addressing how this affects the conclusions and suggesting future strategies for the study.

The clinical applicability of the results is not very clear; I suggest adding a section that explores how these results could be applied in clinical practice, detailing possible strategies and challenges.

Comments on the Quality of English Language

Adequate

Author Response

Thank you for your valuable comments.

Based on your suggestions, we have revised the introduction to emphasize the pathophysiological role of NETosis in various conditions within the Acute Brain Injury (ABI) spectrum, such as Acute Ischemic stroke and Subarachnoid Hemorrhage (lines 70-77).

We appreciated your valuable suggestion to specify why we had chosen to delve into NETosis in ABI instead of other markers. In the discussion (paragraph 4.3 Perspectives on neuroprotection), we stated that “NETosis represents a relatively novel and significant mechanism of immune response that plays a crucial role in the pathophysiology of various acute brain injuries. While other markers have been extensively studied, NETosis offers unique insights into the inflammatory processes, has the advantage of being easily measured at the bedside and has shown to be a promising therapeutic target”.

Regarding the methods, we chose to even include studies with small sample sizes due to the limited number of publications on these topics. We further underlined that sample size was not an exclusion criterion (line 115). However, as mentioned, we excluded studies where NETosis markers were measured in a semi-quantitative manner.

We have also expanded our discussion on the pros and cons of the clinical appraisal tools used for assessing the quality of the collected studies (paragraph 4.7 Limitations of the study).

Lastly, we appreciated your recommendation to integrate preclinical and clinical data. To enhance clarity for readers, we included an illustration to summarise preclinical and clinical findings and possible clinical applicability of these results (Figure 2).

Reviewer 3 Report

Comments and Suggestions for Authors

This study is a REVIEW paper on Net in ABI. This area still has much room for development and the reader may find much of interest. In this regard, we recognize that this is an important paper. In terms of treatment strategies, there is not enough clinical application at this time, but given this potential, it would be good to add a discussion, even if only a few lines, on therapeutic applications in TBI. Please consider this.

Author Response

Thank you for your valuable comments. As you stated, there is no evidence available regarding the impact of targeting NETs on outcome in TBI patients. This is a severe limitation and the subject deserves further study. To enhance clarity for readers, we included an illustration (Figure 2) to  integrate preclinical and clinical data and show possible clinical applicability of the results in all the conditions, including TBI. 

Round 2

Reviewer 2 Report

Comments and Suggestions for Authors

The authors made the suggested modifications and I believe it is suitable for publication.